# Patterns of selection against centrosome amplification in human cell lines

**Marco António Dias Louro**[1]*, **Mónica Bettencourt-Dias**[1], **Claudia Bank**[1,2]

**1** Instituto Gulbenkian de Ciência, Oeiras, Portugal, **2** Institute of Ecology and Evolution, University of Bern, Bern, Switzerland

* mlouro@igc.gulbenkian.pt

## Abstract

The presence of extra centrioles, termed centrosome amplification, is a hallmark of cancer. The distribution of centriole numbers within a cancer cell population appears to be at an equilibrium maintained by centriole overproduction and selection, reminiscent of mutation-selection balance. It is unknown to date if the interaction between centriole overproduction and selection can quantitatively explain the intra- and inter-population heterogeneity in centriole numbers. Here, we define mutation-selection-like models and employ a model selection approach to infer patterns of centriole overproduction and selection in a diverse panel of human cell lines. Surprisingly, we infer strong and uniform selection against any number of extra centrioles in most cell lines. Finally we assess the accuracy and precision of our inference method and find that it increases non-linearly as a function of the number of sampled cells. We discuss the biological implications of our results and how our methodology can inform future experiments.

**Data Availability Statement:** The code developed and used to produce the results of this manuscript can be found in https://gitlab.com/madlouro/centriole-number-variability. All other relevant data

## Author summary

Human cells possess small structures called centrioles, which need to be duplicated and properly segregated to ensure cell viability. Paradoxically, cells with a variable number of excess centrioles are commonly found in cancer. It is thought that these cells arise from centriole overproduction and are subsequently eliminated by selection, such that their frequency is stable in the population. However, it is not known if this overproduction-selection balance is sufficient to explain the observed intra- and inter-population variation in centriole numbers.

In this study, we model the cell population dynamics of abnormal centriole numbers inspired by classical evolutionary theory, and infer overproduction and selection parameters from a panel of 67 human cell lines. Surprisingly, our results indicate that the observed variability in most cell lines can be best explained by models assuming a single penalty against extra centrioles, regardless of their number, and complex overproduction "rules", where multiple centrioles can be gained in a single event. Furthermore, we estimate that selection against extra centrioles is generally very strong. Our work presents a novel quantitative approach to analyse centriole number variation and to further our understanding of the role of centriole number abnormalities in cancer development.

are within the manuscript and its Supporting information files.

**Funding:** This work was supported by EMBO Installation Grant IG4152 and by ERC Starting Grant 804569 – FIT2GO, and FCT (Fundação para a Ciência e Tecnologia) grant PTDC/BIA-BID/32225/2017. M.A.D.L. was funded by FCT research fellowship PD/BD/139217/2018. The funders had no role in study design, data collection and analysis, decision to publish, or preparation of the manuscript.

**Competing interests:** The authors have declared that no competing interests exist.

## Introduction

Centrioles are microtubule-based structures that organise the centrosome and thereby orchestrate microtubule nucleation in vertebrate cells [1, 2]. Centriole number abnormalities are a source of phenotypic heterogeneity in cancer cells. Indeed, centriole numbers show variability both within cancer cell populations and between different cancers [3–5]. The causes and consequences of this heterogeneity are still poorly understood. Moreover, to the best of our knowledge, there exists no quantitative description of how centriole numbers are distributed in cancer cells.

In a proliferating cell population, cells start the cell cycle with two centrioles, which duplicate once and only once during S-phase. After cytokinesis, both daughter cells inherit two centrioles each. This centriole duplication and segregation cycle ensures that centriole number is kept constant across generations [3, 6, 7].

In stark contrast with most proliferating cells, centriole numbers are often de-regulated during cancer development. In particular, cells with abnormally high numbers of centrioles are common in tumors and cancer-derived cell lines, and have been recently identified in preneoplastic tissues [3, 8–12]. Interestingly, within a single population of cancer cells, individual cells often carry different numbers of centrioles. However, the number of centrioles per cell in the population seems to display a specific distribution depending on the cell type [9–12]. The source of this variability within and between cell populations is still poorly understood and calls for the development of quantitative approaches.

The occurrence of extra centrioles, termed centrosome amplification, tends to bear deleterious consequences for the cell by triggering multipolar divisions, cell cycle arrest, and/or by promoting chromosome missegregation [13–16]. Thus, excess centrioles are typically counter-selected and rarely observed in healthy tissues. However, some mechanisms are known to provide protection against centrosome amplification. For example, centrosome clustering mechanisms allow cells to group extra centrioles in two spindle poles, thus improving the viability of daughter cells [13, 14, 16]. Thus, cancer cell lines are generally regarded as being more tolerant to centrosome amplification than normal cells.

Recent data suggest that centriole numbers are maintained at an equilibrium in cell line populations. For instance, it has been observed that after transient centriole elimination, p53-deficient cell populations can seemingly recover their initial distribution of centriole numbers [17, 18]. Similarly, there are reports of extra centrioles being lost over time in cell populations after induction of cytokinesis failure [19, 20]. Since centrosome amplification is typically deleterious for cells, it is likely that centriole numbers in these populations are maintained by a balance of centriole (over)production and negative selection. These dynamics are similar to an evolutionary mutation-selection process, where the *de novo* appearance of deleterious variants in a population is counteracted by natural selection, eventually converging to so-called mutation-selection balance [21].

The dynamics of centriole (over)production and selection are currently unresolved. For instance, extra centrioles may be gained "smoothly" in a dose-dependent fashion through overexpression of key centriole biogenesis regulators, such as Plk4, STIL, and SAS-6 [22–25]. Alternatively, extra centrioles may be gained in sharp transitions—if an otherwise normal cell undergoes cytokinesis failure, it may restart the cell cycle with at least double the normal number of centrioles (as suggested in [26]). Similarly, it is not known if selection strength varies with the number of centrioles. For example, it is possible that centriole clustering is less efficient in resolving multipolar spindles if the cell contains a high number of extra centrioles. In the absence of protective mechanisms, it is possible that the presence of extra centrioles is deleterious, regardless of absolute centriole numbers. Thus, it is not a trivial question how centriole (over)production and selection can generate equilibrium distribution of centriole numbers,

and if these two processes are sufficient to explain the observed centriole number heterogeneity within and between cell populations.

Here, we develop mathematical models of centriole overproduction and selection against centrosome amplification that are predicated on different assumptions on how supernumerary centrioles are produced and how selection operates. We use these models to analyse recently published data on representative cell lines of the progression from Barrett's esophagus to gastroesophageal adenocarcinoma [9] and the NCI-60 panel of cancer cell lines [10]. These two data sets provide us with the opportunity to study how centriole number distributions vary along cancer progression, from pre-malignant to malignant stages, in the case of the Barrett's esophagus data set, and between different cancer types, in the NCI-60 data set.

Employing a model selection approach, we found that models featuring a constant cost of centrosome amplification, irrespective of the number of centrioles in a cell, best explain the empirical distributions for most of the cell lines. Moreover, our results suggest that the distribution of centriole numbers is generally super-exponential, which could be indicative of multi-step centriole number increments. We identified a general trend in the parameter estimates indicating strong selection against extra centrioles but we did not detect significant differences between cell lines. Using simulations, we show that our parameter estimation method is accurate and we predict that its precision increases non-linearly with the number of sampled cells. In summary, our work presents the first quantitative description of how centriole numbers evolve in proliferating cell populations with persistent centrosome amplification and provides a statistical tool for further dissecting the processes that generate within- and between-population variation in centriole numbers.

## Results

### A model of centriole number dynamics in proliferating cell populations

To study how centriole number distributions in proliferating cell populations are generated, we developed a general mathematical model grounded in mutation-selection theory. Our subject of focus is a population of proliferating cells subject to centriole overproduction and selection against extra centrioles. For the purpose of data analysis, we consider that individual cells are in mitosis and fully characterised by their number of extra centrioles, $i$, which can range from zero to an arbitrarily high upper bound, $i_{max}$. Thus, the population can be split by centriole numbers into subpopulations such that, for example, the zeroth subpopulation ($i = 0$) represents all cells containing wild-type centriole numbers (four centrioles). Each subpopulation $i$ has an intrinsic growth rate $r_i$. Centriole overproduction occurs at a rate $\mu_{i,j}$ from subpopulation $i$ to subpopulation $j$, where $j > i$. Thus, we assume that there is no loss of centrioles across cell division (which would be given by a transition from $i$ to $j$ where $j < i$). Centriole overproduction events can be interpreted as gain of $j - i$ centrioles. Since in these cell lines there is a net increase in the number of centrioles compared to the wild-type situation, we make the simplifying assumption that there is no loss of centrioles. Cells that contain fewer than wild-type numbers were rarely observed in the analysed data sets; for simplification purposes, we disregard them in our model and in our analysis. Finally, our model is deterministic and all subpopulation frequencies $P_i$ are continuous variables; i.e., we assume an effectively infinite population size. Taken together, the temporal rate of change in the relative frequency of cells containing $i$ centrioles is given by the following ordinary differential equation:

$$\frac{dP_i}{dt} = \left( r_i - \sum_{j=0}^{i_{max}} r_j P_j(t) \right) P_i(t) + \sum_{k=0}^{i} \mu_{k,i} P_k(t) - \sum_{l=i+1}^{i_{max}} \mu_{i,l} P_i(t) \,, \text{for all } i \leq i_{max}. \tag{1}$$

The dynamics of the population are thus described by a system of $i_{max} + 1$ differential equations. This model is the continuous-time equivalent of the model proposed by Moran [27], in the absence of back mutation, and included in the framework of the original quasispecies model [28–30]. As previously mentioned, several lines of evidence suggest that centriole numbers in proliferating cell populations follow a stable equilibrium distribution when unperturbed [17–20, 31]. We propose the following expression that describes a fully polymorphic equilibrium distribution, i.e. allowing cells of any subpopulation to occur (see S1 Fig for a numerical example and a test of convergence from random initial conditions; see also Materials and methods), for an arbitrary value of $i_{max}$:

$$P_i^* = \frac{\eta_i}{\sum_{j=0}^{i_{max}} \eta_j} \quad , P_0(0) > 0 \tag{2}$$

$$\eta_i = \prod_{j=i+1}^{i_{max}} f(j) \left( \sum_{a_o \in A(i)} \left( \prod_{k \in a_o} f(k) \prod_{m=1}^{|\Phi(i,k)|-1} \mu_{\phi_m, \phi_{m+1}} \right) \right) \tag{3}$$

where

$$f(i) = r_0 - \sum_{j=1}^{i_{max}} \mu_{0,j} - r_i + \sum_{n=i+1}^{i_{max}} \mu_{i,n} \quad \text{for all } i \leq i_{max}, \tag{4}$$

and

$$A(j) = (a_o)_{o \in \mathcal{P}(S^j)} \tag{5}$$

is the sequence containing all elements of the power set $\mathcal{P}(S^j)$. The set $S^j$ is defined as

$$S^j = y \in \mathbb{N} : 0 < y \leq j - 1 \quad . \tag{6}$$

Finally, we define

$$\Phi(j, k) = (\phi_m)_m \in \{\{0\} \bigcup \{a_j \neg k\}\}, \tag{7}$$

with $a_j$ corresponding to the elements of $A(j)$ from the definition in Eq (5).

This set of equations determines the equilibrium balance of centriole overproduction and selection in a cell population, i.e., it determines the predicted proportion of cells with $i$ extra centrioles in an unperturbed cell population, given an arbitrary set of overproduction rates and fitness functions. This general solution allows for the computation of analytical expressions for the equilibrium distributions and their (log-)likelihood under more specific centriole overproduction and selection scenarios, as shown below.

## Distributions of centriole numbers in cell populations tend to be heavy-tailed

Our goal is to infer the balance between selection and centriole overproduction from the shape of the distribution of centriole numbers in samples from 67 cell lines [9, 10]. In these data sets, between 35 and 82 mitotic cells were sampled from a population of cultured cells, and centrioles were identified and counted by co-immunostaining of two centriolar markers.

As a first step, we characterised which type of distribution most likely underlies these data; this is helfpful for identifying more specific models for parameter inference. For example, consider a simple model where extra centrioles are produced at a constant rate $\mu$ and extra centrioles induce a uniform reduced growth rate $r$. This model can be obtained in our general

framework by substituting $\mu_{i,j} = \mu$ for $j = i+1$ and 0 otherwise, and $r_i = r$. Under these assumptions, Eq (2) can be written as

$$P_i^* = \frac{(1 - r - \mu)\mu^i}{(1 - r)^{i+1}}, \quad P_0(0) > 0 \tag{8}$$

It can be readily seen by substitution that this equilibrium is formally equivalent to the geometric distribution:

$$P(X = i) = (1 - p)^i p, \tag{9}$$

where $p$ represents the probability of success in $i$ trials (roughly, the probability of observing an extra centriole in our model). If an analogue to the classical mutation-selection model is sufficient to explain the data, then the data should be geometrically distributed. In contrast, we observed an overrepresentation of cells with high centriole numbers (S2 Fig) compared to a geometric distribution. This is a coarse indication that the distribution of centriole numbers in cell populations is heavy-tailed. If that is the case, then more complex centriole overproduction and/or selection dynamics are required to explain the data.

To test this at the level of individual empirical distributions for each of the 67 cell lines, we fitted geometric and Waring-Yule distributions to each of the empirical population-level distributions of centriole numbers. Here, the Waring-Yule distribution represents a generalised discrete distribution that can potentially account for heavy tails. Then, we calculated the value of the $X^2$ statistic as a measure of goodness-of-fit for both distributions. Finally, we determined the difference between the value of the $X^2$ statistic, $\Delta X^2$, under the geometric and Waring-Yule models, such that positive values indicate a better fit of the Waring-Yule distribution. Conversely, if the geometric distribution is a better fit, we expect that both distributions should converge and yield a $\Delta X^2$ of approximately 0.

Visual inspection of model fits suggests the Waring-Yule (heavy-tailed) distribution is a better fit to the represented empirical distributions (Fig 1A–1C). In addition, our results indicate positive values of $\Delta X^2$ for the majority of cell lines, suggesting a better fit of the Waring-Yule (heavy-tailed) distribution (Fig 1D). For 16 out of 67 cell lines, we obtained values of $\Delta X^2 \approx 0$ (more accurately, $\leq 1$), indicating exponential-like and not heavy tails. Although the control cell lines used in the NCI-60 study rank in the bottom half of cell lines ordered by ascending $\Delta X^2$ value, apart from HaCat, they are not clearly separated from the remaining cell lines. Thus, our results suggest that a simple model reminiscent of classical mutation-selection balance, yielding a geometric distribution of centriole numbers in the population, fails to explain the data for most cell lines. However, it should be noted that some of the 16 cell lines included in the group with near-zero $\Delta X^2$ values contain few cells with centrosome amplification in the sampled population (e.g. OACP4—1 out of 61 cells with centrosome amplification; IGROV1 —1 out of 58 cells with centrosome amplification), in which case there is little information to distinguish between geometric and Waring-Yule distributions.

Ultimately, we are interested in relating the distribution of centriole numbers in the population to the processes that generate it. If the distributions are indeed heavy-tailed, this could be achieved either by weak selection against cells with high centriole numbers or by more complex centriole overproduction mechanisms. For example, since centrioles duplicate in most healthy cells, it is possible that centriole overproduction also occurs in multiples of two (which we will refer to as overduplication). Similarly, after cytokinesis failure, a cell may restart the cell cycle and reduplicate all four centrioles, gaining four extra [26]. Intuitively, overduplication or cytokinesis failure could produce cells with multiples of two and/or four centrioles.

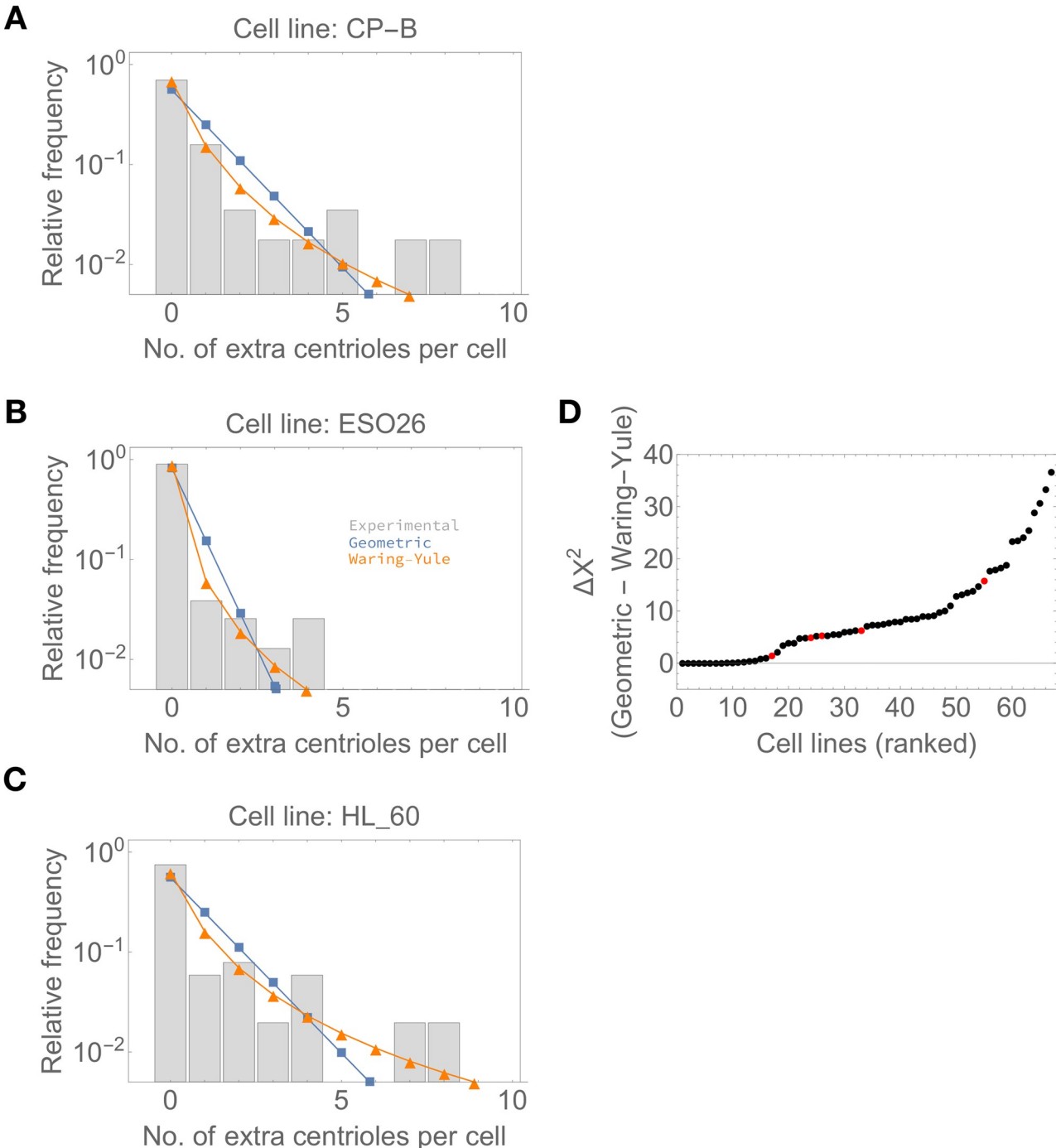

**Fig 1. Distributions of centriole numbers in cell populations tend to be heavy-tailed.** A-C—Examples of empirical distributions for three cell lines (grey bars) and the predicted relative frequencies under geometric (blue) and Waring-Yule distributions (orange), which are representative of distributions with exponential-like and heavy tails, respectively. Number of sampled cells: (A) $n = 57$; (B) $n = 78$; (C) $n = 51$. D—Difference between the calculated $X^2$ value under geometric and Waring-Yule distributions. Cell lines were ranked in ascending order according to the $\Delta X^2$ value. The control cell lines used in [10] are highlighted in red. Higher positive values indicate a better fit of the Waring-Yule distribution to the corresponding empirical distribution, suggesting heavier-than-exponential tails.

Coherently, it can be seen in both individual and pooled distributions that cells with four and eight extra centrioles are particularly abundant in the data (S2 Fig).

In summary, a model parameterised by a single centriole overproduction rate and a single intrinsic growth rate of cells with extra centrioles is not sufficient to explain the data. By visually inspecting the empirical distributions, we hypothesised that including more complex centriole overproduction events in the models, such as centriole overduplication, may provide better fits.

## A candidate set of models based on centriole biology

The general model described in (1) provides a powerful starting point for inferring the distribution of centriole numbers in proliferating cell populations but it is overparameterised with respect to the data under consideration. Moreover, we are interested in comparing different hypotheses regarding specific fitness functions and overproduction parameters that could generate said centriole number distributions. To avoid overfitting and to inspect relevant biological scenarios, we generated 12 candidate models by imposing a set of constraints on centriole overproduction and on selection, based on the previous analysis of the tail of the distributions.

Several cellular processes are known to yield supernumerary centrioles [19, 20, 22–25] but we still lack a quantitative description of their contribution to the generation of cells with extra centrioles. We reasoned that these processes may be parameterised as the rate of gain of a given number of centrioles. As a universal scenario across all models, we considered that extra centrioles can be gained one at a time, at rate $\mu_1$. Since centrioles typically duplicate in number and since we observed an excess of multiples of two and four centrioles in the data, we reasoned that centriole overproduction could also be thought of in terms of extra duplication events. Thus, we considered two additional overproduction "rules", which state two and four extra centrioles can be gained at rates $\mu_2$ and $\mu_4$, respectively.

Similarly, how strongly selection acts depending on the number of extra centrioles remains unknown. We focused on two main possibilities: first, that any abnormal number of centrioles is equally deleterious (resulting in a flat fitness function for all $i > 0$), and second, that the deleterious effect of extra centrioles increases with their number in a cell. Regarding the latter, we assume either an additive or power-law relationship between the number of extra centrioles and intrinsic population growth rate. In all models, we set the intrinsic growth rate for cells with wild-type centriole numbers, $r_0$, to be maximal and equal to 1; i.e., cells that contain no excess centrioles always have maximum fitness. Otherwise, a fully polymorphic equilibrium (i.e. where cells with any number of extra centrioles could, in theory, be observed) would not be reached under these models. The intrinsic growth rates for cells with abnormal centriole numbers ($i > 0$) are defined as (1) $r_i = r$ for all $i > 0$ ("flat" model), (2) $r_i = 1 - \frac{c \cdot i}{i_{max}}$ for all $i > 0$ ("linear" model), and (3) $r_i = 1 - \frac{\log(i+1)}{\lambda \log(i_{max}+1)}$ for all $i > 1$ ("power-law" model). In effect, all these functions represent a fitness landscape in which wild-type-like cells reside on a single fitness peak and all cells with extra centrioles suffer some form of fitness penalty.

The full combinatorial set of the above-fitness functions and "overproduction" rules yields 12 different models, in the following named using the initial of the fitness function and the overproduction steps (Fig 2). For example, F1- - refers to the model featuring a flat fitness function (parameterised by $r$) and single centriole overproduction events (parameterised by $\mu_1$). Note that the models with fewer parameters are nested within the more parameter-heavy models and can be obtained by setting excluded centriole overproduction rates to zero. For instance, F124 yields identical equilibria to F1- - if $\mu_2 = 0$ and $\mu_4 = 0$.

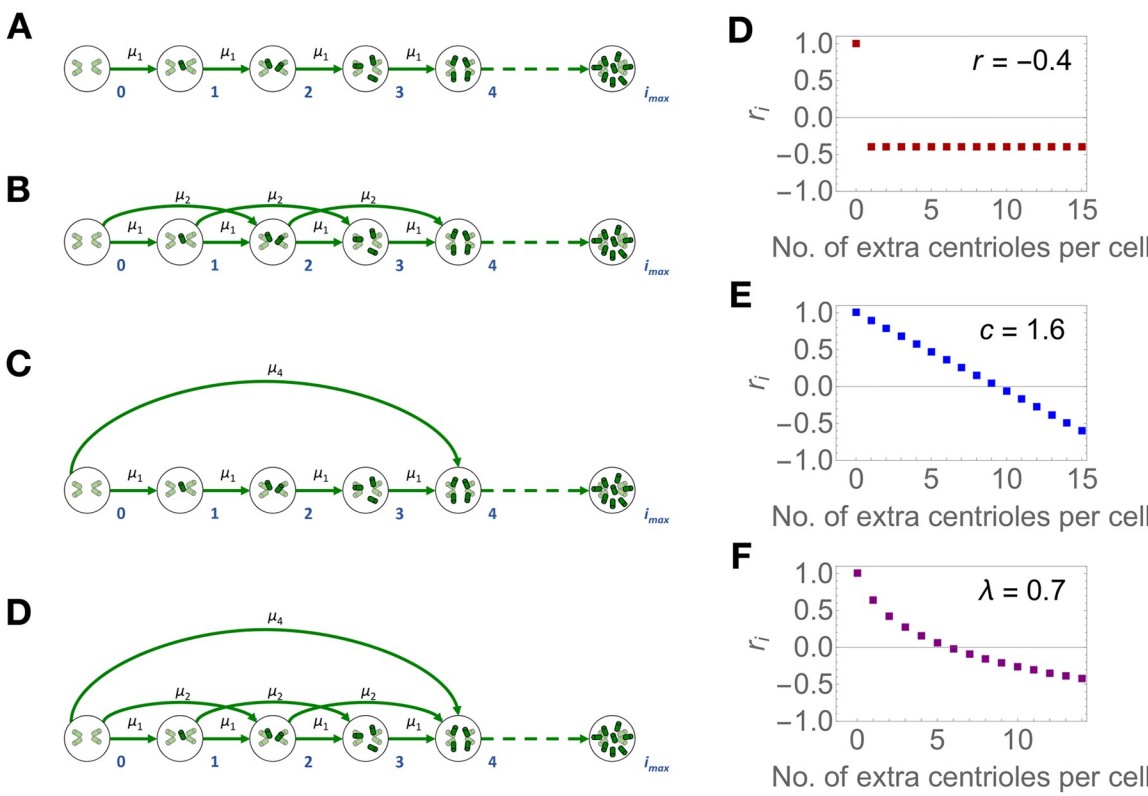

**Fig 2. Centriole overproduction and fitness functions in the candidate models.** A—Single-step centriole overproduction at rate $\mu_1$. B—single- and double-step centriole overproduction events at rates $\mu_1$ and $\mu_2$, C—single- and quadruple-step centriole overproduction events at rates $\mu_1$ and $\mu_4$, D—and all three centriole overproduction events. Circles represent the subpopulation of cells containing $i$ extra centrioles (green cylinders, $i$ indicated by the numbers in blue). Green arrows represent transitions between subpopulations, which correspond to centriole overproduction, occurring at rates $\mu_1$, $\mu_2$ or $\mu_4$. Overproduction events can occur for all $i$ up to $i_{max}$. (E-G) The value of $r_i$ for cells with $i$ centrioles under the (E) flat, (F) linear, and (G) power-law fitness functions, evaluated at the indicated parameter values.

In addition, the 12 models in our candidate set can be obtained by modifying Eq (2) according to the specified centriole overproduction and intrinsic growth rate "rules". In the case of model F124 and by extension all models nested in it simplify to a more concise form that is independent of $i_{max}$ (see supplementary Mathematica Notebook), therefore bypassing the need to assume a potentially artificial upper bound for the number of centrioles per cell.

We note that our focal set of models is neither an exhaustive nor systematic exploration of all possibilities. However, it includes models of different complexity, depending on the number of centriole overproduction parameters. Moreover, it incorporates different biological hypotheses with respect to fitness, and results in both exponential and heavy-tailed equilibrium distributions as described above (see also S3 Fig).

## Models assuming a flat fitness function best explain the data for most cell lines

As a first approach we tested if the most complex models in our candidate set (i.e. the models assuming all three overproduction events) are a good fit to the data. First, we fitted the models to each empirical distribution. Then, we performed a Monte Carlo multinomial test to distinguish between predicted and empirical distributions (Fig 3). We considered that the models were a poor fit if the test yielded a significant $p$-value ($p$-value $\leq 0.05/67$, adjusted according to

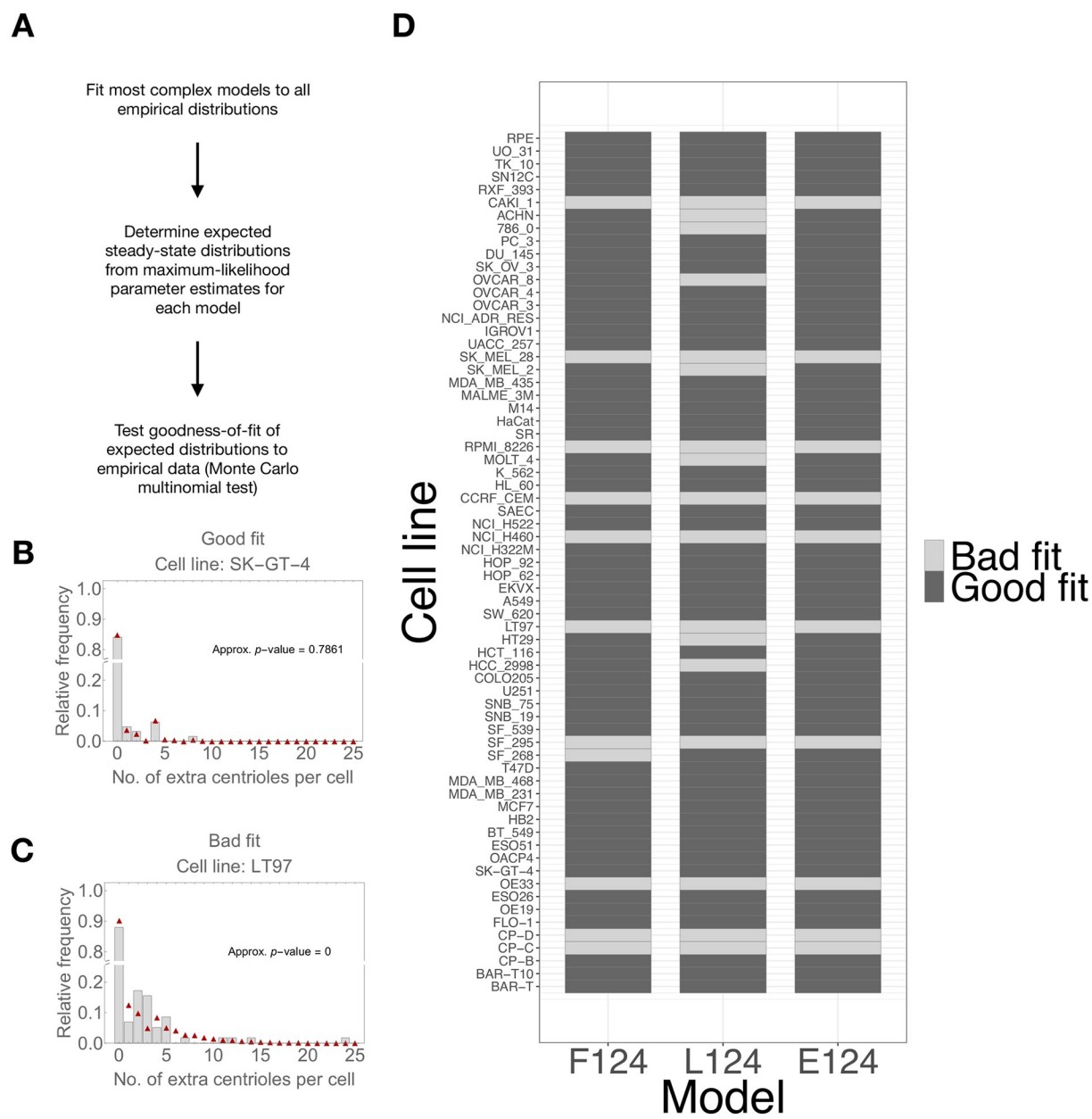

**Fig 3. The most complex models are a good fit to the majority of empirical distributions.** A—Procedure for determining goodness-of-fit. The significance value was set to $\alpha = 0.05$, and adjusted according to the Bonferroni correction for 67 tests. Note that under the null hypothesis, the predicted distribution under a given model is identical to the empirical distribution. B-C—Experimentally observed frequencies of centriole numbers per cell (grey bars) and the predicted frequencies under model F124—flat fitness function, single-, double-, and quadruple-step overproduction parameters (red triangles). The two examples include cases in which we obtained non-significant (good fit) and significant (bad fit) $p$-values. Number of sampled cells: B—$n = 82$. C—$n = 63$. D—Goodness-of-fit of the three most complex models for each cell line.

a Bonferroni correction). Our models showed a good fit for a majority (56 for the "flat" model, 52 for the "linear" model, 57 for the "power-law" model) of cell lines.

Thus, we concluded that the most complex models are a good fit to the data. For 10 out of 67 cell lines, all three models were a poor fit to the corresponding empirical distributions. These cell lines tended to display some proportion of cells with extremely high centriole

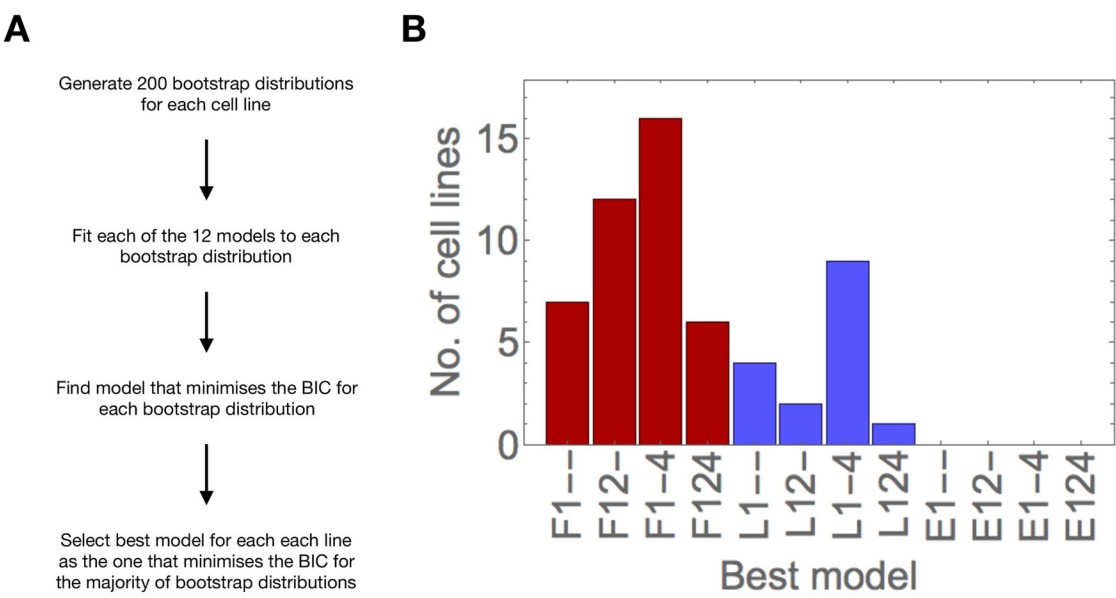

**Fig 4. Models assuming a flat fitness function best explain the data for most cell lines.** A—Procedure for model selection. B—Models with a flat fitness function explain the data for most cell lines. Number of cell lines for which the corresponding model was selected as the best. The fitness function of the models is indicated in red (flat) or blue (linear).

numbers ($\geq 15$), which are very rare under all our models. We took a conservative approach and excluded these cell lines from further analysis.

Next, we asked which of the 12 models best explains the observed data. We generated 200 bootstrap distributions for each cell line by drawing a random sample with replacement from each empirical distribution, and fitted each of the 12 models. Then, we calculated the Bayesian Information Criterion (BIC) score from the resulting maximum log-likelihood value. The model that minimised the BIC for the largest number of bootstrap distributions was selected as the best for each cell line (Fig 4). Strikingly, the best models for 41 out of 57 cell lines assumed the flat fitness function, including all six lung and kidney cell lines in the NCI-60 panel, and the two metaplasia and one dysplasia cell lines in the Barrett's esophagus data set (see S4 Fig for model selection results grouped by tissue of origin). 16 cell lines are best explained by models assuming the linear fitness function; these cell lines are not associated to any specific tissue types or developmental stages. No cell line is best explained by models with a power-law fitness function. In contrast, there is more variability with respect to the best set of centriole overproduction parameters.

Next we analysed the number of bootstrap distributions that was selected for each cell line (S5 Fig). We observed that the best models for each cell line were selected for a maximum of 198 (99%) and a minimum of 40 (20%) bootstrap distributions, with a median of 110.5 (55.25%) bootstrap distributions (S5(B) Fig). In addition, in some cases, the BIC score was equal for models assuming the flat and power-law fitness functions (see supporting code). This means that the decision for the most appropriate model is sometimes not clear, which can be either due to the models yielding indistinguishable distributions, or the data not being sufficiently informative to distinguish between models (see also below).

Then we examined models sharing the same fitness function as the best model. In total, the same fitness function was selected for a maximum of 199 (99.5%) and a minimum of 78 bootstrap distributions (39%), with a median of 150 (75%) (S5(C) Fig). Therefore, the same fitness function was selected more consistently than individual models.

Despite some uncertainty in model selection, we concluded that most empirical distributions are best explained by models assuming a flat fitness function, whereas a smaller subset of cell lines was best explained by models with a linear fitness function.

### Parameter estimates indicate strong selection against excess centrioles

We next tested if we could distinguish between different cell lines based on their parameter estimates. Following our model selection results, we focused on model F124, which includes a constant fitness function and allows for all possible overproduction rules. First, we tested the sensitivity of the fitting as a function of the parameters by evaluating the log-likelihood expression at values around the maximum, and looked for correlations between pairs of parameters. We observed that all parameters are uncorrelated for model F124, and thus can be independently estimated from the data (S6 Fig).

Second, we analysed the maximum-likelihood $r$ and $\mu_1$ estimates obtained for the previously generated non-parametric bootstrap distributions, under model F124 (Fig 5). We observed globally negative median estimates of the intrinsic growth rate $r$, indicating strong selection against cells with extra centrioles. The median estimates for the single-step overproduction rate $\mu_1$ were relatively low but more variable than those of the intrinsic growth rate $r$. Indeed, for most cell lines, the median estimate of $\mu_1$ fell within the range of 0-0.423, with two cell lines scoring over 0.7 (SE268 and HT19). However, the confidence intervals for both parameters were considerably wide, spanning almost the whole parameter range in the case of the intrinsic growth rate $r$, such that we could not identify significant differences between cell lines or by tissue of origin. Thus, our results indicate a pattern of low to moderate centriole overproduction rates and intrinsic growth rates for cells with extra centrioles.

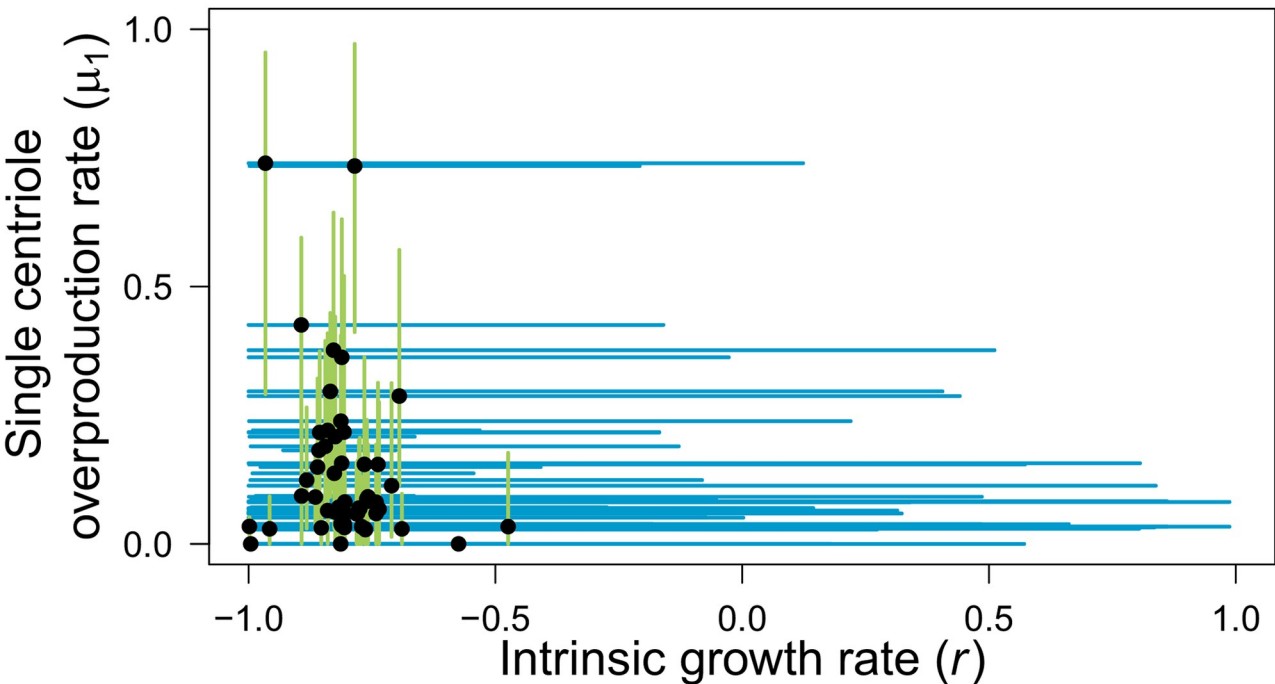

**Fig 5. Parameter estimates indicate strong selection against excess centrioles but there is considerable uncertainty in parameter estimation.** Median estimates (black dots) of $r$ and $\mu_1$ obtained from 200 non-parametric bootstrap distributions for each empirical distribution. Lines indicate 95% confidence intervals (0.025 to 0.975 inter-quantile distance) for the intrinsic growth rate $r$ (blue) and the single-step overproduction rate $\mu_1$ (light green).

Next, we addressed the accuracy and precision of our inference method. If our method is accurate, parameter estimates from model simulations should converge to the input parameter values. In addition, we expected that errors in parameter estimation should be similar between model simulations and the ones obtained in the non-parametric bootstrap, for samples of the same size. To test this, we performed parametric bootstrapping, where we resampled the expected distributions under model F124 instead of resampling the empirical distributions as in the non-parametric bootstrap. As input values, we used the previously calculated median $r$, $\mu_1$, $\mu_2$, and $\mu_4$ values from the 200 non-parametric bootstrap distributions for each cell line. We assumed a sample size, i.e. number of sampled cells, equal to that obtained in the corresponding data set and generated 200 parametric bootstrap distributions. Then, we fitted model F124 to each parametric distribution and analysed the estimated parameter values.

Our results show that the median parameter estimate obtained from the parametric mostly agrees with the input value (S7 Fig), indicating that our method is accurate. Moreover, confidence interval lengths were similar to those obtained from the non-parametric bootstrap distributions. For some cell lines, the errors obtained for the intrinsic growth rate $r$ differed between the non-parametric and parametric bootstrap distributions. This is likely due to the lower sensitivity of the maximum likelihood values to changes in $r$ compared to $\mu_1$ (S6 Fig).

We concluded that parameter estimation accuracy and precision are similar when data is simulated either from the empirical or predicted distributions. Since the confidence interval length is influenced by sample size and we did not detect systematic biases in our inference method, it is likely that the precision of our parameter estimates mainly depended on the number of sampled cells per cell line.

## Accuracy and precision of the inference method increase non-linearly with the number of sampled cells

To provide statistical guidance for future experiments, we asked how much the precision of the parameter estimates and accuracy of model selection could be improved by increasing the sample size, within experimentally feasible limits. To address this question, we reasoned that the estimated values from the median expected distribution would provide a useful test case. We identified the median expected distribution by measuring the Euclidean distance between the vector whose elements are the four estimated parameter values for a given bootstrap distribution to the vector of the lowest possible values for each parameter (equating to $[-1, 0, 0, 0]$ for $r$, $\mu_1$, $\mu_2$, and $\mu_4$, respectively).

We used the parameter values corresponding to the median Euclidean distance as input for model F124 and simulated 200 parametric bootstrap distributions in a range of sample sizes. For comparison purposes, we assumed a sample size of 35 and 83, which correspond respectively to the lowest and the highest number of cells obtained experimentally in our data sets. In addition, we considered realistic sample sizes of 50, 100, 150, 200, and 250 cells. Finally, we simulated parametric bootstrap distributions with a sample size of 1000 to analyse the properties of our inference method if larger sample sizes were attainable.

Subsequently, we fitted the three most complex models. We counted the number of times each model maximised the log-likelihood function out of the 200 bootstrap distributions for each simulated sample size. Note that the BIC is unnecessary because the three models have the same number of parameters. In addition, we analysed the parameter estimates of model F124 from the simulated distributions.

Since we performed simulations under model F124, this model should fit best most of the bootstrap distributions. It is also trivial that errors in parameter estimation will become smaller

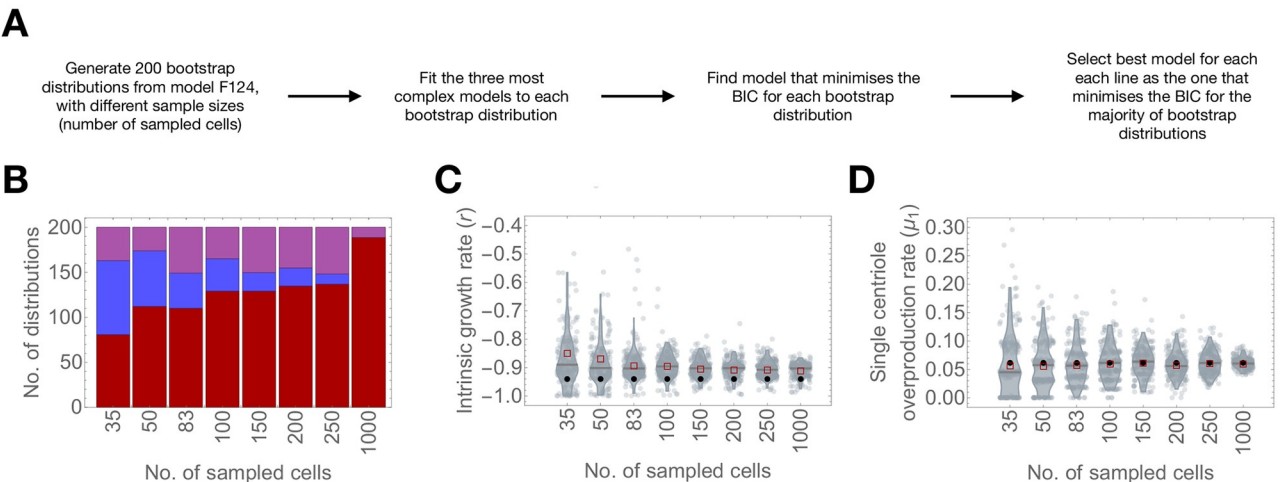

**Fig 6. Accuracy and precision of the inference method increase non-linearly with the number of sampled cells.** A—Number of bootstrap distributions for which model F124 (red), L124 (blue), and E124 (purple) fitness functions were selected as the best in simulations of model F124 as a function of bootstrap sample size. Input parameter values for the simulations: $r = -0.940$, $\mu_1 = 0.062$, $\mu_2 = 0.096$, $\mu_4 = 0.288$. B-C—Bootstrap distribution of parameter estimates for intrinsic growth rate $r$ (B) and single-step overproduction rate $\mu_1$ (C) as a function of bootstrap sample size. Black dots indicate the median estimated value and the green dot indicates the input value used in the simulations.

for larger sample sizes. Regardless, we were interested in quantifying how often model F124 is identified as the best model and how confidence interval length varies with the number of sampled cells.

Fig 6 shows the results for model selection from simulated data. We observed that within the range of experimentally obtained sample sizes, model F124 is correctly identified in 81 (40.5%), 112 (56%), and 110 (55%), out of 200 parametric bootstrap distributions, for a sample size of 35, 50, and 83 simulated cells, respectively. Model F124 is only marginally outperformed by model L124 for a sample size of 35 (selected for 81 and 82 bootstrap distributions, respectively). Nevertheless, we observed that the number of bootstrap distributions for which the true model is selected increases to 129 (59.5%) for a sample size of 100, and to 137 for a sample size of 250 (67.5%). For the maximum sample size tested, the true model was selected for 189 (94.5%) bootstrap distributions.

Next, we inspected how the distributions of parameter estimates vary with sample size. We observed that the input value always falls within the confidence interval but we noted that the median intrinsic growth rate $r$ was consistently overestimated with respect to the input value (Fig 6C). Nevertheless, parameter estimation was fairly accurate for higher sample sizes. We also observed a near two-fold decrease in confidence interval length between the minimum (35) and maximum (83) experimentally obtained sample sizes, and a further 1.55, 1.91, 2.02 and 2.01-fold decrease between the maximum experimentally obtained sample size (83) and examples containing 100, 150, 200, and 250 simulated cells, respectively. For 1000 simulated cells, we obtained confidence intervals with a length of 0.109, corresponding to a 2.43-fold decrease compared to the maximum experimentally obtained sample size.

Unlike for intrinsic growth rates, we obtained extremely accurate median estimates for $\mu_1$ regardless of sample size (Fig 6D). Thus, it is possible that the overestimation of the intrinsic growth rate $r$ is because the estimated values lie close to the lower bound (-1), and not an inconsistency generated by the model. We observed that the confidence interval for the single-step overproduction rate $\mu_1$ decreased from 0.194 to 0.138 (1.41-fold decrease), for sample sizes of 35 and 83, respectively, and further to 0.114 for a sample size of 100 (1.21-fold decrease

compared to a sample size of 83) and 0.083 for a sample size of 250 (1.66-fold decrease compared to a sample size of 83). For 1000 simulated cells, confidence interval lengths were as low as 0.0412.

In conclusion, both model selection accuracy and parameter estimation precision increased non-linearly with sample size. Importantly, we predict that increasing the number of sampled cells within a feasible range (between 100 and 200) can greatly improve our inference power. However, it should be stated that these results may change depending on the number of cells with centrosome amplification. For example, if there are few abnormal cells in the population, it is probably harder to distinguish between models. Conversely, if cells with extra centrioles are more frequent, it is expected that models become easier to distinguish. In addition, the range of parameter values should be taken into account to avoid estimation biases, such that estimated values do not fall close to the bounds.

## Discussion

We here combined analytical and statistical methods to characterise abnormal centriole number distributions in populations of human cell lines. Adopting classical mutation-selection balance theory from population genetics, we developed a set of mathematical models for analysing a broad panel of cell lines, which are representative of the diversity along cancer development and between different cancer types. Using a model selection approach, we found that a constant and heavy cost of excess centriole numbers is a common feature of the best approximating models for the majority of cell lines. In addition, we quantified how uncertainty in the model selection and parameter estimation procedures can be reduced by obtaining larger sample sizes in the future. We show that integrating statistical information into experimental setups could reveal potential differences between cell lines in the mechanisms that cause abnormal centriole number distributions. Importantly, our population-level approach recognises and quantifies the variation in centriole numbers that has recently been observed in experimental data.

### Dynamics of centriole numbers in proliferating cell populations

To the best of our knowledge, we provide the first quantitative description of centriole number dynamics in populations of proliferating cells. Past studies, both experimental and theoretical, on the population-level response to supernumerary centrioles investigated how wild-type numbers are recovered after perturbation [17, 18], and also highlighted the role of negative selection in driving this process. These previous studies have mainly focused on distinguishing cells with wild-type centriole/centrosome numbers from cells with abnormal numbers. Here, we explored the full centriole number variation that has been observed in experimental studies. We described the heterogeneity in centriole numbers per cell within and between cell populations, and we evaluated which type of underlying fitness function is most likely to generate the observed variation within the population. Interestingly, our analysis suggests that selection acts strongly against any number of excess centrioles in most cell lines. This means that deleterious effects arise as soon as excess centrioles are produced, whereas the actual number does not seem to matter for selection, and that the shape of the distribution is determined chiefly by the mechanism(s) of centriole overproduction.

### Implications for the biology of centrosomes

Our results show that the centriole number distribution within a population carries important biological information. First, as we argued above, we inferred a constant and heavy cost of abnormal centriole numbers. Whereas understanding variation has long been a staple of

evolutionary studies, it has often been overlooked in cell biology. However, on a similar subject, it has been reported that different mechanisms of organelle biosynthesis, such as *de novo* assembly or fusion, display specific signatures that can be identified by relating the mean and the variance in their distributions [32]. Thus, valuable insight can be gained from a broader quantitative description of the data.

Second, simple models incorporating single-centriole overproduction events and a constant fitness function (i.e. akin to the classical formulation of mutation-selection balance in population genetics and widely explored in quasispecies models as reviewed in [33]) were sufficient to explain the shape of the centriole distribution in a few cell lines, whereas in others a more complex relationship between selection and overproduction improved the fitting. In the latter case, our analysis indicates that the shape of the distribution depends more on how supernumerary centrioles are acquired rather than on how they are eliminated—i.e. model fits are less sensitive to changes in intrinsic growth rates than to the mode of overproduction. This raises the hypothesis that various cellular mechanisms might lead to overproduction whereas selection "punishes" the presence of any number of excess centrioles.

The biological processes associated with our postulated single-, double- and quadruple-step centriole overproduction events may be entirely different. For example, overproduction of two centrioles could occur due to centriole re-duplication after premature disengagement [34] in wild-type cells. Quadruple overproduction could be a consequence of cytokinesis failure followed by reduplication of all four centrioles [26].

We identified models with a flat fitness function as the best for most cell lines, suggesting that centrosome amplification *per se* is deleterious, regardless of the number of extra centrioles. Intuitively, one could expect that a higher number of centrioles would induce stronger selection because the mechanisms that eliminate cells with centrosome amplification seem to involve some form of "counting". For example, one of the main sources of cell death in cells with extra centrioles is multipolar divisions [13, 14]. It is possible that it is harder for cells to cluster extra centrioles if there are more of them. Thus, one could expect that the probability that a cell undergoes a multipolar division increases with the number of extra centrioles. Likewise, it has been recently proposed that a molecular complex called the PIDDosome triggers p53-dependent cell cycle arrest by "counting" excess mother centrioles [15]. Thus, it seems likely that the efficacy of the PIDDosome in detecting extra centrioles should increase with the number of extra centrioles. If the probability of multipolar divisions and PIDDosome-dependent cell cycle arrest increases with the number of extra centrioles, this could imply a linearly decreasing fitness function as the one we inferred for some cell lines. However, it is still not clear how these or other mechanisms respond to the number of extra centrioles.

We anticipate that future experimental work will address these mechanisms in greater detail, upon which our models can be refined to integrate mechanistic details of overproduction rather than the current general overproduction rates. That could, in turn, allow for a more specific statistical inference of when and how centriole overproduction occurs in different cell lines. Ideally, our modeling and inference approach will eventually link experimental information about centriole distributions with genomic inference of cancer-line and -stage specific molecular alterations.

## Limitations of this study

The observation that cells with extra centrioles are relatively rare in the analysed data sets is a major determinant of the uncertainty of our model selection and parameter estimation procedures. However, we showed that a modest increase in the number of analysed cells can potentially mitigate these issues. In addition, mapping model parameters to the biological system

may not be trivial. Importantly, one of the aspects we simplified is how extra centrioles are allocated to the daughter cells after mitosis, or centriole segregation. Whereas after correct centriole duplication, each daughter cell inherits a single centriolar pair, it is still unclear how extra centrioles are distributed. However, while this manuscript was in review, Sala et al. reported that the degree of asymmetry in centriole segregation increases with the number of extra centrioles in RPE-1 cells transiently overexpressing Plk4 [31]. Since most cells with centrosome amplification in the samples we have analysed contain only a few extra centrioles, it can be expected that centriole segregation is predominantly symmetrical. Moreover, this study reports that extra centrioles are capable of duplication. Thus, cells should have the same number of centrioles as their mothers on average. In light of these results, we conclude that our deterministic approximation is reasonable.

Furthermore, the fact that cells in the data sets are mitotic is convenient for the inference of centriole overproduction and selection parameters because allowed us to eliminate confounding variables related to the cell cycle. First, it means that centriole number variations along the cell cycle, which would be expected in an asynchronously dividing population, need not be considered. Second, it allowed us to disregard differences in cell cycle progression within and between cell lines. Altogether, characterising centriole number distributions along cell cycle progression is an entirely different problem. From a purely theoretical standpoint, it requires a more detailed implementation of the timing of centriole duplication and the length of each cell cycle phase. This would be desirable in the broader context of characterising centriole number distributions in proliferating cell populations but is beyond the scope of our current analysis.

## Towards unraveling the causes and consequences of centriole number changes in cancer

Numeric aberrations of centrioles and their putative link to cancer formation have long been described [3], although accurate quantifications of centriole numbers in tumor biopsies and cancer-derived cell lines have emerged only recently [9–12]. To this day, the contribution of centrosomal anomalies to cancer development remains controversial, with some studies showing that higher numbers, via Plk4 overexpression, can initiate or aggravate tumorigenesis [35, 36], and others showing that it is not sufficient and may even slow down progression [37, 38]. On the other hand, extra centrioles are associated with other cancer hallmarks, such as aneuploidy [14, 39] and invasion [40–42], and often correlate with a more aggressive cancer phenotype [3, 7]. The widespread occurrence of centriole number abnormalities in a cancer setting makes them attractive as prospective biomarkers and as therapeutic targets [43]. Thus, understanding the underlying causes of these abnormalities is important for biomedical research.

Here, we adopted evolutionary theory to quantify the variation in centriole numbers within cancer cell populations. That is because ultimately, centriole number abnormalities are highly heterogeneous both within and between cancer cell populations. In order to predict how these abnormalities evolve during cancer development, and how they may interweave with other cancer hallmarks, it is crucial to have a quantitative understanding of how extra centrioles emerge in these cells, and how the cells cope with them. For example, in the case of the Barrett's esophagus progression model, the increase in centriole numbers from the metaplasia to the dysplasia stages can be explained by loss of p53 [9]. This can be interpreted as a reduction in the strength of negative selection, since p53 can lead to cell-cycle arrest or cell death in the presence of extra centrioles. If this was true, it would be interesting to quantify how strong the decrease in selective pressure and if it is sufficient, by itself, to account for the shift in centriole number distributions.

We showed that, pending an increase in statistical power, our modelling framework can be used to infer these changes and further our understanding of the relationship between extra centriole numbers and cancer development.

## Materials and methods

### Experimental data

For our analysis, we considered two recently published data sets. The first data set corresponds to 13 cell lines derived from different tissues in the Barrett's esophagus cancer progression model. These include pre-malignant (metaplasia and dysplasia) and malignant stages (adeno-carcinoma and lymph node metastasis). The second data set corresponds to 53 cell lines from the NCI-60 panel, a group of cell lines that spans multiple cancer types (leukemia, melanoma and lung, colon, brain, ovary, breast, prostate, and kidney, cancers), as well as five non-cancerous cell lines. In both cases, the data correspond to centriole number counts in mitotic cells. Out of the 71 cell lines, four were discarded from the analysis: for three of the cell lines, we could not compute the equilibrium expression, due to high $i_{max}$); for the remaining cell line, no cells with centrosome amplification were recorded, under which conditions the models can be trivially solved by setting $\mu_1$, and/or $\mu_2$, and/or $\mu_4$ to zero. As previously stated, we do not take into account any cell with less than four centrioles (which represent approximately 1.9% of the total). Additional experimental details can be found in the corresponding publications [9, 10].

### Mathematical analysis

Eq (1) describes the rate of change in the equilibrium frequency of the subpopulation of cells containing $i$ centrioles. To obtain the equilibrium solution in Eq (2), we solved $\frac{dP_i}{dt} = 0$ for all $i$, for $i_{max} = \{3, 4, 5\}$, using Wolfram Mathematica, and proposed a general expression for increasing $i_{max}$. Then, we verified if the expression was correct by comparing it to the steady-state obtained from numerical integration of Eq (1), and further confirmed that the system converges to the same equilibrium for random initial conditions (S1 Fig).

To ensure Eq (2) yields exclusively non-negative values for all $i$, we added the following constraint:

$$\mu_{i,j} > 0 \qquad \wedge \qquad r_0 > \sum_{j=0}^{i_{max}} \mu_{0,j} + r_i - \sum_{k=i}^{i_{max}} \mu_{i,k}. \tag{10}$$

The first term indicates that centriole overproduction rates $\mu_{i,j}$ are strictly positive, otherwise higher centriole numbers would not be reachable. The second term indicates that the intrinsic growth rate of wild-type cells must exceed the sum of the intrinsic growth rates of cells with $i$ extra centrioles and the rate at which their frequency increases as a function of centriole overproduction. Breaking this constraint would lead to the depletion of cells with wild-type centriole numbers, which is not observed in any of the data in our analysis.

### Model fitting and selection

To fit the models, we first derived a general (log-)likelihood expression:

$$\ln \mathcal{L}(\theta_M | p_i) = \sum_{i=0}^{i_{max}} p_i \ln \left( P_i^* | \theta_M \right) \tag{11}$$

where $\theta$ is the tuple of parameters in model M, $P_i$ is the equilibrium solution derived in (2), $p_i$

**Table 1. Model parameters.** The indicated ranges were used as constraints when estimating parameters.

| Parameter | Range |
|---|---|
| $r$ | $]-1, 1[$ |
| $c$ | $]0, 2[$ |
| $\lambda$ | $]0, 2[$ |
| $\mu_1$ | $[0, 1[$ |
| $\mu_2$ | $[0, 1[$ |
| $\mu_4$ | $[0, 1[$ |

is the observed relative frequency of cells containing $i$ centrioles, and $i_{max}$ indicates the sub-population of cells harboring the maximum observed number of centrioles. For ease of comparison, we assumed $i_{max}$ to be the maximum overall number of extra centrioles per cell in the data (30). We fitted the models by numerical maximisation of the log-likelihood function, according to model parameters and the indicated range of values (Table 1).

The Bayesian Information Criterion (BIC) was calculated according to:

$$BIC = \ln(n)\kappa - 2\ln\hat{\mathcal{L}}(\theta_M|p_i), \tag{12}$$

where $\hat{\mathcal{L}}$ is the maximum likelihood estimator for a given model, $\kappa$ is the number of parameters in the model and $n$ is the sample size (number of cells observed in a given cell line). The BIC accounts for sample size and the number of model parameters, such that more complex models are penalised. When compared to another frequently used model selection criterion, the Akaike Information Criterion (AIC), the added sample size penalty is useful given that the number of sampled cells per cell line is limited. We selected the best model for each cell line by finding the one that minimises the BIC score. We performed model fitting and selection using Wolfram Mathematica. The results for model selection based on the empirical distributions alone (as opposed to the bootstrap samples) were confirmed in Python to check for numerical inconsistencies in log-likelihood values.

## Bootstrapping

Non-parametric bootstrap samples were generated by drawing data points, with replacement, from the empirical distribution for each cell line. The sample size for each bootstrap distribution was equal to the one obtained experimentally. Parametric bootstrap samples were generated by drawing, with replacement, from each predicted distribution. In other words, we fitted the models to the empirical distributions in question to obtain an expected distribution under the respective models, and sampled from this distribution.

## Statistical analyses and data visualisation

After obtaining expected distributions under geometric and Waring-Yule models, using Wolfram Mathematica built-in functions, we calculated the value of the $X^2$ statistic for each pair of empirical and expected distributions. It should be noted that the $X^2$ statistic has limited power for analysing the data at hand, such that we used it only for comparative purposes.

When testing goodness-of-fit of the most complex candidate models, we first determined the expected distribution under each of the models, for each cell line, and then performed Monte Carlo multinomial tests between the expected distribution and the corresponding empirical distribution. The Monte Carlo step consisted of 10,000,000 simulations under the expected distribution. We determined (approximated) $p$-values based on the proportions of

simulations that yielded more extreme results than the data. The significance level was set to 0.05, and adjusted according to the Bonferroni correction for multiple testing (one for each of the 67 cell lines).

All plots were produced in Wolfram Mathematica or R. Also see the supplementary code in the accompanying repository: https://gitlab.com/madlouro/centriole-number-variability.

## Supporting information

**S1 Fig. The solutions of the system of ordinary differential equations converge to the expected equilibrium value.** A—Comparison between numeric integration of the general model and the corresponding equilibrium expression (2), evaluated at the same parameter values. We assumed $i_{max}$ = 15 and generated pseudo-random parameter values for all $r_i$ and $\mu_{i,j}$. B—Comparison between numeric integration of the general model from different initial conditions and the corresponding equilibrium expression. We generated a set of pseudo-random parameter values for all $r_i$ and $\mu_{i,j}$ and initial conditions. Note that the y-axis is in log-scale. (TIF)

**S2 Fig. Cells with high number of centrioles occur frequently across all data sets.** Best fitting geometric distribution (blue) to the pooled distribution of centriole numbers in all sampled populations. Number of sampled cells: $n$ = 3746. (TIF)

**S3 Fig. Models in the candidate set can produce both geometric and heavy-tailed distributions.** Best fitting geometric (blue) and Waring-Yule (orange) distributions to 1,000,000 simulated data points (relative frequencies indicated as grey bars). A—Data simulated from model F1-- ($r$ = −0.3, $\mu_1$ = 0.6). B—Data simulated from model F124 ($r$ = −0.5, $\mu_1$ = 0.2, $\mu_2$ = 0.1, $\mu_4$ = 0.1). Data points were generated by multinomial sampling from the equilibrium distributions evaluated at the indicated parameter values. Note that the y-axis is in log-scale. (TIF)

**S4 Fig. Best fitness functions per tissue type.** Models sharing the fitness function of the best model for each cell line in the data sets. A—Barrett's esophagus data set, grouped by developmental stage. B—NCI-60 data set, grouped by tissue type (including cancer and non-cancer cell lines). "Other" refers to an RPE cell line that was used as a control. (TIF)

**S5 Fig. Bootstrap support of the candidate models.** A—Number of bootstrap distributions (out of the 200 generated for each cell line) explained by each model. We obtained indistinguishable BIC scores for multiple models in 381 bootstrap distributions spread across 26 different cell lines ("Mult."). C—Number of bootstrap distributions explained by models sharing the same fitness function as the best model for each cell line. The fitness functions of the models are indicated in red (flat), blue (linear), purple (power-law), and green (multiple models). (TIF)

**S6 Fig. Model parameters are not correlated and the likelihood value shows little sensitivity to $r$.** We fitted the model to a random empirical distribution from the analysed data sets and calculated the likelihood values centered around the maximum as a function of pairwise combinations of parameter values. A—$r$ and $\mu_1$; B—$r$ and $\mu_2$; C—$r$ and $\mu_4$; D—$\mu_1$ and $\mu_2$; E—$\mu_1$ and $\mu_4$; F—$\mu_2$ and $\mu_4$. Note that the scale for $r$ is different from that of the centriole overproduction parameters $\mu_1, \mu_2$, and $\mu_4$. (TIF)

**S7 Fig. Parameter estimation is accurate and estimation errors are likely due to small sample sizes.** Difference between the median value of the parametric bootstrap distribution for each parameter value and the input value for the simulated data (in black) and difference between the confidence interval length of the non-parametric and parametric bootstrap distributions (in yellow). A—$r$. B—$\mu_1$. C—$\mu_2$. D—$\mu_4$.
(TIF)

## Acknowledgments

We would like to acknowledge Carla A. M. Lopes and Gaëlle Marteil for in-depth discussions of their work with the Barrett's esophagus and NCI-60 cell lines, and Telmo Cunha for his work on a prior version of the models. We thank all members of the Bank and Bettencourt-Dias labs for their insights and critical reading of this manuscript. We would like to acknowledge various participants of the "From Molecular Basis to Predictability and Control of Evolution" workshop at the Nordita Institute in Sweden for inspiring discussions.

## Author Contributions

**Conceptualization:** Marco António Dias Louro, Mónica Bettencourt-Dias, Claudia Bank.

**Data curation:** Marco António Dias Louro.

**Formal analysis:** Marco António Dias Louro.

**Funding acquisition:** Claudia Bank.

**Investigation:** Marco António Dias Louro.

**Methodology:** Marco António Dias Louro, Claudia Bank.

**Project administration:** Mónica Bettencourt-Dias, Claudia Bank.

**Resources:** Mónica Bettencourt-Dias, Claudia Bank.

**Software:** Marco António Dias Louro.

**Supervision:** Mónica Bettencourt-Dias, Claudia Bank.

**Validation:** Marco António Dias Louro, Claudia Bank.

**Visualization:** Marco António Dias Louro.

**Writing – original draft:** Marco António Dias Louro, Claudia Bank.

**Writing – review & editing:** Marco António Dias Louro, Mónica Bettencourt-Dias, Claudia Bank.

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
