## [Decision Letter · Decision Letter 0]

7 Sep 2020

Dear Mr, Dias Louro,

Thank you very much for submitting your manuscript "Patterns of selection against centrosome amplification in human cell lines" for consideration at PLOS Computational Biology.

As with all papers reviewed by the journal, your manuscript was reviewed by members of the editorial board and by several independent reviewers. In light of the reviews (below this email), we would like to invite the resubmission of a significantly-revised version that takes into account the reviewers' comments.

We cannot make any decision about publication until we have seen the revised manuscript and your response to the reviewers' comments. Your revised manuscript is also likely to be sent to reviewers for further evaluation.

Sincerely,

Jacopo Grilli

Associate Editor

PLOS Computational Biology

Douglas Lauffenburger

Deputy Editor

PLOS Computational Biology

Reviewer's Responses to Questions

**Comments to the Authors:**

Reviewer #1: In this manuscript Marco et al., have collected centriole amplification data from two previous publications that quantitated the number of centrioles in various cell lines. Based on the number of centriole distribution authors have tried to fit models that could best explain the centriole distribution in the individual cell line population. Based on their model (and its obvious too) authors state that there is selection against centriole amplification irrespective of their origin. The methods are in detail and manuscript is very well written. Below are my suggestions to improve this manuscript further.

The abstract starts with “The presence of extra centrioles, or centrosome amplification, is a hallmark of cancer.”. We know that cancerous cells have uncontrolled division/growth. The manuscript would read better by including discussion on “how cancer cell evades the bias against centriole amplification”?

In NCI 60 cell line panel there are many cell lines originating from solid tumors as well as non-solid tumors (though a handful). Was there any difference in centriole amplification distribution between cell lines derived from solid and non-solid tumors? Though I do see one example in Figure 1.

In Figure 1D one fourth of the total cell lines do not show any difference between geometric and Waring-Yule distribution. On the other hand, one fourth of cell lines show relatively very high (ranges 10 to ~40) delta chi square values. What authors have to say about this? What are those cell lines at the two extreme ends? Please discuss about the features of these two groups of cell lines (if any) in the relevant section. Also, where are the five non cancer cell lines on this graph. Do the non-cancerous cell lines show different pattern? If the non-cancerous cell lines show similar pattern like cancerous cell lines then this should be mentioned or shown as supplementary figure.

In Figure S2 any underrepresentation of wild-type-like cell is not evident. The statement should be modified.

There is no discussion or any effort on splitting the data based on their tissue of origin, at least for those tissue types that have more than 5 cell lines to see if there is any significant difference between the tissue types.

For Figure S3 please include more detail in the methods section, how exactly simulation was done.

In the discussion section authors state that “Interestingly, our analysis suggests that selection acts strongly against any number of excess centrioles in most cell lines. This means that deleterious effects arise as soon as excess centrioles are produced, whereas the actual number does not seem to matter for selection.”. If the actual number of centrioles does not matter then they should see the same frequency across all additional number (compared to WT) of centrioles. However, based on their result this is not the case. Please explain.

Reviewer #2: Review of Patterns of selection against centrosome amplification in human cell lines, by Dias Louro et al., submitted to PLOS Computational Biology

This interesting paper uses model generation and fitting to data to determine how extra centrioles are produced and selected against in cancer and pre-cancerous cells. The topic is biomedically important, and the authors’ approach, on the whole, very well reasoned and carefully laid out. The choice of a deterministic formalism is commendable. The variables in it, if need be, can be interpreted as frequentist probabilities. It is interesting to see novel, distinct mutation-selection distributions arising in this instance of extra-genetic (in the narrow sense of DNA) inheritance. The model can serve as an example for mutation-selection of cell structural features that are inherited, with modification, through cell division. Although the present model is consciously constructed in such a simplified way as not to dwell on the fitness and mutability consequences of the still incompletely understood cytoskeletal mechanisms that involve supernumerary centrioles, centriole number dynamics should still be a grateful subject for more mechanistically detailed modeling in the future, compared with other potential applications of this approach to cell structure inheritance. The conclusions reached here through sophisticated comparative model fitting – that there is a flat and low fitness landscape outside the basal centriole number – are not hard to accept, given what we know about abnormal mitoses. Notwithstanding, they are far from trivial and add to this still developing field. I have two minor remarks, and one major one.

1 (major). Does the model presuppose that unequal division of centrioles between daughter cells never happens? Unequal division, which does occur (the paper refers to the relevant literature in the introduction), results in both an increase and a decrease of the centriole numbers in the progeny. Both are, in general, heritable. I believe that unequal division of the centriole number is essentially the norm past one extra centriole duplication. The Discussion section on limitations of this study speaks of “centriole segregation” as having been neglected; it appears the authors may be referring to the issue I am raising here. However, the meaning of segregation is somewhat obscure in that passage. If it is indeed argued that strong negative selection makes unequal distribution between daughter cells inconsequential, it would make equally inconsequential all mutations in the centriole number, other than those from the basal number. It seems that the model behavior that is evaluated against empirical data will be rather strongly affected by unequal centriole distribution between daughter cells.

2 (minor). The steady-state solution is said to be locally stable. The methods suggest that this is an observation based on convergence from randomly sampled states. How do we know it is not globally stable? In the absence of analysis, stating simply that the model converges to this solution from randomized initial conditions may be preferable, although this is normally taken as evidence of uniqueness and global stability of the solution. It seems to me that the model, based on its form, should have only one time-independent solution, absent exotic parameter and initial condition choices.

3 (minor). Mutation-selection literature references might help the reader where this approach is first mentioned. Similarly, the reader will be helped by an explicit literature reference to the original quasi-species model (for primordial biopolymer) that had the discussed type of flat fitness landscape.

**Have all data underlying the figures and results presented in the manuscript been provided?**

Reviewer #1: Yes

Reviewer #2: None

PLOS authors have the option to publish the peer review history of their article (what does this mean?). If published, this will include your full peer review and any attached files.

Reviewer #1: **Yes: **Pankaj Kumar

Reviewer #2: **Yes: **Ivan Maly
---

## [Decision Letter · Decision Letter 1]

3 Feb 2021

Dear Mr, Dias Louro,

We are pleased to inform you that your manuscript 'Patterns of selection against centrosome amplification in human cell lines' has been provisionally accepted for publication in PLOS Computational Biology.

Best regards,

Jacopo Grilli

Associate Editor

PLOS Computational Biology

Douglas Lauffenburger

Deputy Editor

PLOS Computational Biology

Reviewer's Responses to Questions

**Comments to the Authors:**

Reviewer #1: Thank you for answering all the comments and incorporating the suggested changes. The manuscript should be accepted now.

Reviewer #2: My suggestions have been addressed.

**Have all data underlying the figures and results presented in the manuscript been provided?**

Reviewer #1: Yes

Reviewer #2: None

PLOS authors have the option to publish the peer review history of their article (what does this mean?). If published, this will include your full peer review and any attached files.

Reviewer #1: **Yes: **Pankaj Kumar

Reviewer #2: **Yes: **Ivan Maly

---

## [Editor Report · Acceptance letter]

10 May 2021

PCOMPBIOL-D-20-00383R1 

Patterns of selection against centrosome amplification in human cell lines

Dear Dr Dias Louro,

I am pleased to inform you that your manuscript has been formally accepted for publication in PLOS Computational Biology. Your manuscript is now with our production department and you will be notified of the publication date in due course.

With kind regards,

Andrea Szabo
